# Online Video Anomaly Detection

**DOI:** 10.3390/s23177442

**Published:** 2023-08-26

**Authors:** Yuxing Zhang, Jinchen Song, Yuehan Jiang, Hongjun Li

**Affiliations:** 1School of Information Science and Technology, Nantong University, Nantong 226019, China; 2State Key Lab. for Novel Software Technology, Nanjing University, Nanjing 210023, China

**Keywords:** video surveillance, real time, online video anomaly detection

## Abstract

With the popularity of video surveillance technology, people are paying more and more attention to how to detect abnormal states or events in videos in time. Therefore, real-time, automatic and accurate detection of abnormal events has become the main goal of video-based surveillance systems. To achieve this goal, many researchers have conducted in-depth research on online video anomaly detection. This paper presents the background of the research in this field and briefly explains the research methods of offline video anomaly detection. Then, we sort out and classify the research methods of online video anomaly detection and expound on the basic ideas and characteristics of each method. In addition, we summarize the datasets commonly used in online video anomaly detection and compare and analyze the performance of the current mainstream algorithms according to the evaluation criteria of each dataset. Finally, we summarize the future trends in the field of online video anomaly detection.

## 1. Introduction

Video surveillance systems are widely used in shopping malls, hospitals, banks, streets, smart cities and other places to enhance the safety of public life and property [1]. However, the massive video data generated by the rapid development of video surveillance systems have brought great challenges to abnormal event detection based on manual interpretation. The traditional way of finding anomalies by manually viewing the monitored image records not only consumes a lot of human resources, but also may cause errors or omissions that cannot be remedied in time [2]. Therefore, it is very important to develop a technology that can automatically analyze and find abnormal conditions in surveillance videos without relying on a lot of manpower, which is video anomaly detection technology.

Video anomaly detection is the task of identifying abnormal events from video sequences that reflect frames that are significantly different from normal events. A video anomaly is usually an abnormal appearance or motion attribute in a video [3]. Due to the scarcity and diversity of abnormal samples, video anomaly detection methods usually only model the normal sample distribution. In the detection process, video frames or video clips that deviate from the normal sample distribution are regarded as anomalies [4]. In video surveillance scenarios, common examples of anomalies are pedestrians driving on sidewalks, pedestrians walking on lawns and vehicles incorrectly parking, as shown in Figure 1 [5].

On the one hand, online video anomaly detection usually refers to actively learning frames of normal events in a video sequence in the system or model, constantly updating the definition of a normal event. On the other hand, it refers to reducing the processing time of each incoming video frame online for fast spatio-temporal localization of abnormal events and real-time video anomaly detection. Ideal video anomaly detection is defined as the process of detecting and tracking abnormal events in real time [6]. However, most video anomaly detection methods are still offline. Considering that video anomaly detection can play an important role in ensuring security and confidentiality and preventing potential disasters, video anomaly detection systems should have real-time decision-making capabilities. For example, traffic accidents, robberies, fires in remote areas and other events need to be detected and solved in time through real-time detection of abnormal events [7]. However, the research focusing on online and real-time detection methods is very limited. Therefore, online (real-time) video anomaly detection has important research significance.

In summary, this paper briefly introduces the offline video anomaly detection method, mainly analyzes and studies the online video anomaly detection based on sample annotation and the technical methods used, and further classifies the methods on this basis, as shown in Figure 2. At the same time, the mainstream datasets of online video anomaly detection are sorted out, and the development direction of online video anomaly detection is prospected.

## 2. Common Datasets and Assessment Criteria

### 2.1. Common Single-Scene Anomaly Detection Datasets

Table 1 lists the specific information of current scene datasets commonly used for online video anomaly detection.

(1)UMN dataset. The UMN dataset [8] has 11 videos containing three different scenes: campus lawn, indoor and square. This dataset belongs to the global abnormal behavior dataset.(2)UCSD Pedestrian dataset. The UCSD Pedestrian dataset [9] was captured on a university campus pavement. The current frame-level anomaly detection accuracies achieved on the UCSD Ped1 and UCSD Ped2 datasets are 97.4% and 97.8%, respectively. Although the accuracy rates are relatively high, this dataset is still one of the few more popular benchmark datasets due to its relatively adequate number of anomalies and anomaly types.(3)CUHK Avenue dataset. The CUHK Avenue dataset [10] contains 16 training videos and 21 test videos with a total of 47 abnormal events, including throwing objects, loitering and running. The size of the people changes depending on the position and angle of the camera.(4)ShanghaiTech (Shanghai, China) dataset. The ShanghaiTech dataset [11] contains 13 scenes with complex lighting conditions and camera angles, containing 130 abnormal events and over 270,000 training frames. In addition, pixel-level annotations of the abnormal events are given.(5)UCF-Crime dataset. The UCF-Crime dataset [12] is a large dataset of 128 h of video. It consists of 1900 continuous unedited surveillance videos containing a total of 13 real-life aberrant behaviors, namely abuse, arrest, arson, assault, road accident, burglary, explosion, combat, robbery, shooting, theft, shoplifting and vandalism.(6)The DOTA dataset. The DOTA dataset [13] is the first traffic anomaly dataset to provide detailed spatio-temporal annotation of abnormal objects. It contains 4677 videos, and each video contains exactly one abnormal event. The anomalies consider seven common traffic participant categories, namely pedestrians, cars, trucks, buses, motorbikes, bicycles and riders.

### 2.2. Evaluation Criteria

There are two main criteria for evaluating the effectiveness of anomaly detection: (1) the receiver operating characteristic (ROC) and its corresponding area under the curve (AUC) and (2) the equal error rate (EER). There are two levels of AUC evaluation criteria: the pixel level and the frame level. The frame-level criterion determines the correctness of anomaly detection on a frame-by-frame basis and only determines whether a frame contains an anomaly, without the need to accurately locate the spatial location of the anomaly. The pixel-level criterion requires the spatial location of the anomaly to be accurately located, and the detection is deemed successful when the detected anomaly area overlaps with the real anomaly area by 40%. For the evaluation criteria of EER, some literature argues that abnormal events are rare in real life and there is a serious sample imbalance between normal and abnormal events, so using EER as a metric can be misleading in practical applications [14,15].

## 3. Offline Video Anomaly Detection Methods

Depending on the data and the method, learning methods for anomaly detection can be broadly classified into supervised, unsupervised, weakly supervised and self-supervised learning. They can also be divided into reconstruction-based and prediction-based video anomaly detection methods according to deep learning methods.

In offline video anomaly detection methods, the learning methods of anomaly detection can be roughly divided into supervised learning, unsupervised learning, weakly supervised learning and self-supervised learning. Among them, the generative cooperative learning method is a truly unsupervised anomaly detection algorithm based on unlabeled training videos [16]. Implicit dual-path automatic encoding and a normal feature distribution model based on a normalized flow generation model can be used for anomaly detection [17]. Three-dimensional convolutional auto-encoders can be trained for anomaly detection by denoising reconstruction errors and adversarial learning strategies, which can better distinguish normal and abnormal events even without any supervised information [18]. Weakly supervised video anomaly detection methods can capture abnormal events at different scales from local to global spatio-temporal perception, and they can identify abnormal segments from the entire video segment marked as abnormal [19]. A multi-scale convolutional neural network (CNN) can be captured from video using relation-aware feature extractors, which is a new weakly supervised method [20]. Video anomaly detection methods based on self-supervised learning generally model and generate feature-level anomalies by using offline learning dictionaries and self-supervised learning [21]. A self-supervised learning method can solve a multi-label fine-grained classification problem [22], and CutPaste self-supervised learning can be used to improve and train the network to better adapt to the data [23].

State-of-the-art video anomaly detection methods use deep learning techniques and are divided into reconstruction-based and prediction-based video anomaly detection methods. Among them, the cascade reconstruction model combined with the frame reconstruction network and the optical flow prediction network can be used to increase the reconstruction error of video anomalies [24], so as to better distinguish normal and abnormal events. A generative assistant discriminative network is used to distinguish normal future frames from noisy future frames by introducing a helper, and this method can better learn discriminative features to distinguish small differences between normal and abnormal events [25]. Other methods use convolutional auto-encoder architectures to address large reconstruction errors caused by abnormal events with irregular appearance or motion behavior [26], and noise modulates the end-to-end training of generative adversarial networks to regulate the generalization ability of the reconstruction network [27]. The encoder and decoder are used to generate reconstruction errors to separate abnormal events from normal events to detect abnormal events [28]; in order to alleviate the loss caused by feature compression, the authors of [29] proposed the skip attention gate mechanism to complete the spatio-temporal information fusion between the encoder and the decoder. Patch-wise prediction method bidirectional architecture [30] combines the prediction error of appearance and motion and the commit error between original features and memory items to calculate the anomaly score and determine whether a frame is abnormal [31]. An encoder based on a 3D CNN and a decoder based on a 2D CNN constitute a network with spatio-temporal consistency enhancement. This network uses the difference in prediction quality between normal and abnormal events to infer whether an anomaly occurs [32]. Methods such as the adaptive Markov jumping particle filter [33] and active-Langevin model for predicting abnormal behavior have also been used for video anomaly detection [34].

Although most video-based abnormal event detection methods work well in simple scenes, they often suffer from low detection rates and high false alarm rates in complex motion scenes. Therefore, video anomaly detection methods should shift from offline detection to online (real-time) detection, so as to find abnormal events in time and make appropriate decisions. So, real-time abnormal event detection in video streams is a worthy future research direction.

## 4. Online Video Anomaly Detection Methods

In order to solve the problem of detecting and identifying anomalies in video accurately and in real time, this section mainly summarizes an overview of online video anomaly detection methods based on sample annotation, technical approaches and other methods and further subclassifies them to obtain a clearer understanding.

### 4.1. Online Anomaly Detection Based on Sample Annotation

#### 4.1.1. Online Anomaly Detection Based on Supervised Learning

Supervised learning is the process of mapping all data samples to different class labels through training a model with known data samples and their one-to-one corresponding labels. However, due to the scarcity of abnormal videos, online video anomaly detection methods based on supervised learning are relatively rare. As shown in Table 2, to address the challenges and limitations of anomaly detection localization in real-time surveillance, a multi-objective neural anomaly detection framework has been proposed [35]. This method combined a future frame detector based on generative adversarial networks and a lightweight object detector (Yolov3). The extracted features were used for online anomaly detection by nonparametric statistical algorithms. Intelligent traffic monitoring systems need to be able to detect anomalies such as traffic accidents in real time, and [36] adopted the Yolo object detection framework and used a more heuristic approach to detect anomalies. Methods included background estimation, road mask extraction and adaptive thresholding to remove some erroneous anomalies. Finally, the final anomaly was detected and analyzed using a decision tree. In order to ensure the online working mode and fast response of autonomous driving anomaly detection, in [37], researchers used the video swin transformer model to model short-term memory through the current frame and a few previous frames, used temporal correlation for behavior detection, introduced the LSTM module to model the past long-term memory, and obtained the overall context information from the video in real-time and online scenarios. Luo et al. [38] jointly used features extracted by spatially optimized temporal coherent sparse coding and features extracted by time-stacked recurrent neural network auto-encoders in real-time anomaly detection. Meanwhile, the distance between sparse codes was weighted by the similarity between adjacent frames.

Although these online anomaly detection methods based on supervised learning can detect the labeled types of anomalies, their performance deteriorates sharply in complex scenes. Since there are many types of anomalies and they cannot all be manually labeled, anomaly detection approaches turn to unsupervised learning methods.

#### 4.1.2. Online Anomaly Detection Based on Unsupervised Learning

Unsupervised learning does not need labeling of the data, which not only saves costs but makes the learned features more adaptive and richer. Therefore, unsupervised learning methods are widely used in online anomaly detection. As shown in Table 2, Chaker et al. [39] used a window-based method to construct a spatio-temporal cuboid; modeled crowd behavior through local social networks; and realized online abnormal behavior detection, localization, and global updating. Dang et al. [40] proposed a method for unsupervised anomaly detection, map coding and anomaly-aware path planning using aerial robots. This method uses deep learning features and support vector machines combined with Bayesian fusion on a map for multi-view anomaly consensus. Real-time detection of abnormal events in video streams is a difficult challenge due to the volatility of anomaly definitions, time constraints and the adaptability of parametric models. To address the problem, ref. [41] used deep learning features for image classification and used Bayesian technology to fuse, encode and update the abnormal information on the occupancy map reconstructed in real time, so as to realize the detection and location of environmental anomalies for robots. In addition, a path-planning method was proposed to maximize the sensor observation entropy of the automatic exploration location area and abnormal area. In [13], authors introduced a new large-scale benchmark dataset detection of traffic anomaly (DOTA) and proposed a new video anomaly detection evaluation metric called the spatio-temporal area under the curve (STAUC). Researchers predicted the future position of an object in a short period of time by using a multi-stream RNN framework. Most existing methods used manually labeled data and defined parameters to train offline detectors and have difficulty modeling in changing scenes. An online self-organizing map (SOM) was proposed and achieved more satisfactory results for local anomaly detection in outdoor crowded scenes [12]. Compared with the original SOM [42], the online SOM model has the ability to adjust the learning rate and domain size according to scene changes. In order to avoid the poor performance of the data space caused by the zero neural structure far away from the actual sample in SOM, Ref. [43] adopted a more flexible neural network—an online growth neural network. The model achieved anomaly detection in the changing data space by self-using parameter learning operations online, estimating the learning efficiency and adaptively adjusting the network size. To some extent, it effectively reduced false alarms and missed detection caused by model aging and threshold aging in unstable environments. The incremental spatio-temporal learner in [44] updated and distinguished new abnormal and normal behaviors in real time through active learning with fuzzy aggregation for anomaly detection and localization in real-time surveillance video streams. In addition, Monakhov et al. [45] proposed an algorithm for anomaly detection in complex videos such as surveillance video. The algorithm is based on the HTM architecture of a grid, which simplifies data by segmentation technology and realizes real-time unsupervised anomaly detection. It has good anti-noise ability and online learning ability and can combat concept drift when performing real-time anomaly detection in video surveillance. The authors of [46] studied an anomaly detection problem for a video dataset taken by drones in parking lots. These researchers proposed a new anomaly detection technique using multiple feature extraction methods, such as GoogleNet, HOG, PCA-HOG and HOG3D, to classify the extracted features into two categories, namely abnormal and normal, and constructed four one-class support vector machine (OCSVM)-based models for anomaly event detection. The goal of this research was to enable automated abnormal event detection to assist UAV surveillance tasks. Different from the above methods, the GridNet method proposed in [47] utilizes deep auto-encoders for anomaly detection under image-agnostic conditions. By using the meta-representation of the scene image and introducing a multi-part loss function consisting of an adjustable variational loss term and a new loss term, namely the Soft-F1 loss term, the proposed method can accurately detect object anomalies in a scene image.

These online anomaly detection methods based on unsupervised learning can reduce the need for manual annotation resources, but the performance is poor in complex scenarios. One of the main difficulties faced by video analysis is the huge amount of data, and only a small number of videos contain important information. Therefore, developing algorithms to automatically detect abnormal events in streaming or archived videos can significantly improve the efficiency of video detection. This is also the focus of future research.

#### 4.1.3. Online Anomaly Detection Based on Weakly Supervised Learning

Semi-supervised learning is an important type of weakly supervised learning that can improve the learning effect by using labeled and unlabeled data without fully supervised tasks. As shown in Table 2, Ref. [48] introduced an auto-encoder framework based on multi-modal semi-supervised deep learning convolution–bidirectional long short-term memory model (CNN-BiLSTM). The proposed method generates video clips by segmenting the online video stream from RGB + D sensors into fixed-size video frames, and then it uses a CNN-BiLSTM auto-encoder semi-supervised framework that only utilizes weakly labeled video samples for training and is able to perform anomaly detection without abnormal video samples. In addition, a framework for multivariate, data-driven and sequential detection of anomalies in high-dimensional systems based on the availability of labeled data was proposed in [49], which is suitable for both semi-supervised and supervised environments. These methods provide an effective framework for semi-supervised learning, which can be applied to various fields to improve the efficiency and accuracy of machine learning.

By using a large number of coarsely labeled samples, weakly supervised learning can use data more effectively than semi-supervised learning to improve the performance of the model. However, video anomaly detection based on weak supervision has difficulty in distinguishing abnormal and normal events in complex situations during training, which leads to non-optimal separation boundaries. To solve this problem, some authors extracted features from a pre-trained 3D CNN [50] and considered the local spatio-temporal differences by introducing a dissimilarity attention module. A ranking loss function considering the detection score of the dissimilarity attention module (DAM) and the temporal attention weight of the DAM for gradient calculation was used for optimization. To enhance the detection of abnormal segments in video streams, a transformer-based multi-scale long–short-term context model for online anomaly localization was proposed [51]. The locator is able to recover useful information from historical observations and compensate for future semantics by anticipation, providing critical context for detecting what is happening. However, this method required considerable computational cost. To address issues such as the lack of ability to model the temporal relationship between video clips and the inability to extract enough discriminative features to distinguish between normal and abnormal clips, a weakly supervised temporal discrimination algorithm was proposed by Huang et al. A transformer-styled temporal feature aggregator (TTFA) was used to aggregate temporal features in [52], which addressed the issue of temporal relationship modeling and feature discrimination. Additionally, a self-guided discriminative feature encoder (SDFE) was introduced to enhance the separability of features. The performance of temporal modeling and anomaly detection in video clips can be improved with the application of these methods. Online video anomaly detection based on weak supervision is an important research topic in the field of intelligent surveillance. Previous research has tended to use a unified single-stage framework that struggles to address both online constraints and weakly supervised settings. Online video anomaly detection using a two-stage- or multi-stage-based framework is a worthwhile research direction for the future.

**Table 2 sensors-23-07442-t002:** Summary of online anomaly detection methods based on sample labeling.

Reference	Method	Architecture	Remarks
Doshi et al. [35]	Supervised learning	GAN and YOLOv3	A multi-objective neural anomaly detection framework was proposed.
Aboah et al. [36]		YOLO and decision-making tree	The Yolo object detection framework was adopted, and the final anomaly was detected and analyzed using a decision tree.
Rossi et al. [37]		Transformer and LSTM	The video swin transformer model was used, and the LSTM module was introduced to model long short-term memory.
Luo et al. [38]		Sparse-coding-inspired DNN	Using spatially optimized temporally coherent sparse coding and temporally stacked recurrent neural network auto-encoders.
Chaker et al. [39]	Unsupervised learning	Social network model	The detection, location and global update of online abnormal behavior were realized.
Dang et al. [40]		AlexNet and one-class SVM + Bayesian encoding	Deep learning features and Bayesian technology were used to detect and locate environmental anomalies, and a path-planning method was proposed.
Zhao et al. [41]		Dynamic sparse coding	Online sparse reconstruction methods and online reconfigurability were proposed to detect abnormal events in videos.
Yao et al. [13]		Two-stream RNNS	A new dataset and evaluation metric STAUC were introduced to perform online anomaly detection.
Sultani et al. [12]		Deep multiple-instance learning	A MIL algorithm with sparsity and smoothness constraints was proposed.
Harada et al. [43]		Online growing neural gas	The anomaly detection of the changing data space was realized.
Nawaratne et al. [44]		Incremental spatio-temporal learner	It updated and distinguished new abnormal and normal behaviors in real time and was used for anomaly detection and location in monitoring.
Monakhov et al. [45]		GridHTM	The visual analysis method based on the extended HTM algorithm was studied, and the data were simplified by the segmentation technique.
Chriki et al. [46]		One-class support fvector machine	Four different features were extracted with GoogleNet, HOG, PCA-HOG and HOG3D for automated abnormal event detection to form four OCSVM models.
Bozcan et al. [47]		GridNet	Achieved anomaly detection that was insensitive to vision-related issues and proposed a novel loss function based on grid representation.
Khaire et al. [48]	Weakly supervised learning	CNN-BiLSTM	The proposed CNN-BiLSTM auto-encoder framework implements semi-supervised anomaly detection.
Mozaffari et al. [49]		KNN	Presenting a framework for multivariate, data-driven and sequential detection of anomalies in high-dimensional systems based on the availability of labeled data.
Majhi et al. [50]		Dissimilarity attention module	A dissimilarity attention module with local context information was introduced into anomaly detection for real-time applications.
Liu et al. [51]		TPP and OAL	Accurate segment-level pseudo-label generation and future semantic prediction were achieved through a two-stage framework of “decoupling and parsing”.
Huang et al. [52]		TTFA and SDFE	Proposing a transformer-style TTFA and a boot SDFE.

### 4.2. Online Anomaly Detection Based on Technical Methods

From the perspective of technical methods, online anomaly detection methods can be divided into two aspects: those based on deep learning and those based on transfer learning.

#### 4.2.1. Online Anomaly Detection Based on Deep Learning

Due to the excellent performance of deep learning, many deep-learning-based frameworks are available, and these are compatible with many platforms to enable complex problems to be solved. The current mainstream method is to identify patterns of abnormal behavior through deep learning. As shown in Table 3, traditional online anomaly detection solutions are limited to operating on videos with only a few abnormal frames. In order to break this limitation, the perceptron proposed in [53] was optimized online to reconstruct video frames pixel by pixel from their frequency information. Based on the movement of information between adjacent frames, the incremental learner updated the parameters of the multi-layer perceptron after observing each frame, thus allowing the detection of abnormal events along the video stream. An incremental learner was also utilized to online optimize the multi-layer perceptron to produce results that detect abnormal events along the video stream. Ata-Ur-Rehman et al. [54] proposed a new particle prediction model and a weighted likelihood model to accurately detect abnormal video frames and abnormal regions in video frames. At the same time, the above two models can effectively extract and utilize information in the form of size, motion and unknown features from video frames. Currently, no fully automatic monitoring techniques can simultaneously detect and interpret abnormal events, and most require human intervention to identify the nature of abnormal events and select appropriate countermeasures. To solve this problem, in [55], a classification model was used for anomaly detection and compressed to efficiently accommodate resource-constrained devices. Additionally, abnormal activities were effectively classified using a visual transformer with a sequential learning model and attention for anomaly recognition. Due to the data-intensive end-to-end training of neural networks, the feature representation extracted from the spatio-temporal features of video was not interpretable. In [56], the authors proposed an interpretable framework with zero-shot cross-domain adaptability. The asymptotic optimality of the used sequence detector was derived in terms of minimizing the average detection delay in the maximal sense while controlling the false alarm rate. A new architecture based on deep learning was provided for the detection of violent acts in videos in [57]. The architecture consisted mainly of 3D DenseNet, a multi-head self-attention layer, and a bidirectional convolutional LSTM module. The detection of abnormal behavior was achieved by encoding the relevant spatio-temporal features. With the design of this architecture, the spatio-temporal information in the video was effectively captured, and the accuracy and reliability of abnormal behavior detection were improved. In [58], researchers proposed the first important surveillance scene dataset considering indoor and outdoor environments and adopted CNN-based optical flow estimation for the first time. It represented motion in industrial surveillance video by adding residual layer blocks encoded by a temporal optical flow feature (TOFF) and used Mask R-CNN to segment instances of important objects in the scene. To address the challenge of complex abnormal behavior analysis, a system that starts with anomaly detection was introduced in [59]. The image of the pedestrian in the video is transmitted to the abnormal behavior analysis module through TCP/IP communication. Subsequently, real-time recognition of behaviors is performed using 3DResNet, enabling the analysis and recognition of complex abnormal behaviors. In [60], researchers proposed a lightweight framework containing convolutional neural networks, trained on input frames obtained in a computationally cost-effective method, which effectively described and distinguished between normal and abnormal events. Due to the insufficient availability of real-world datasets, the first comprehensive abnormal event generation system capable of simulating typical specific abnormal events was developed in [61]. By combining 3D CNNs and non-local mechanisms, the performance of anomaly video detection on synthetic datasets was improved. A recurrent 3D GAN was proposed for adaptation to reduce the domain gap, enabling the conversion of synthetic data into realistic video sequences. In order to realize the automatic detection of abnormal behavior in an examination room, the authors [62] studied an improved algorithm based on yolov3, which used the K-means algorithm, Giouloss algorithm, Focal loss algorithm and Darknet32 algorithm. In addition, The frame-alternate dual-thread method was used to optimize the detection process. Direkoglu et al. [63] first proposed a new concept of multiplying the angle difference by the optical flow amplitude calculated in the current frame to form the motion information image (MII). The proposed motion information image system (MIIS) can well distinguish normal and abnormal events. The proposal of the MII model and its combination with a CNN was a new method for global abnormal crowd behavior detection.

The false alarm rate is a key metric for evaluating the performance of online video anomaly detection. However, there are relatively few online anomaly detection methods based on deep learning in the current methods, so it is worthwhile to establish an online anomaly detection framework from the perspective of deep learning. At the same time, the timely location of abnormal events is also the development direction of future research. This involves not only the detection and location of the abnormal event itself, but also the accurate detection of the occurrence time of the abnormal event. Therefore, one of the future research directions is to comprehensively consider the time factor to achieve the precise location of abnormal events in anomaly detection. In summary, establishing an online anomaly detection framework from the perspective of deep learning and taking the time factor into account to achieve accurate detection and localization of abnormal events is a worthy direction for future research and development.

#### 4.2.2. Online Anomaly Detection Based on Transfer Learning

The currently proposed methods and models only perform well on some publicly available datasets but often do not perform well when used in real-world complex video surveillance. As shown in Table 3, in order to produce models with high generalization, some scholars have used transfer learning methods to improve them. In [64], the k-nearest neighbor method was employed to remove misclassified fixed objects, and then K-means clustering was used to identify potential abnormal regions. Finally, the occurrence time of the anomaly was determined by the backtracking anomaly detection algorithm, which calculated the similarity statistics. To address the reason why online decision making was neglected in the field of surveillance video, some researchers have proposed a hybrid use of transfer learning via neural networks and statistical k-nearest neighbor (KNN) decision-making methods to detect video anomalies forever and in an online manner with limited training. For video anomaly detection, the authors of several papers significantly reduced the training complexity by leveraging transfer learning, while presenting a new framework for statistical arbitrary shot sequence anomaly detection, which was able to continuously learn from very few samples. A continuous learning algorithm was also proposed, which consisted of a feature extraction module based on transfer learning and a statistical anomaly detection module that made online decisions on decision rules and continuously updated them [65,66]. To overcome the challenges of lack of modularity, cross-domain adaptability, interpretability and real-time anomaly event detection in existing video anomaly detection, the authors of [67] proposed the first multi-task learning framework capable of cross-adaptation, fewshot learning and continuous learning for limited-data video anomaly detection, while presenting the first semantic embedded-based approach for video anomaly detection using deep metric learning. In addition to a single modular framework, a plug-and-play architecture that includes transfer learning was introduced in [68]. It included an interpretable transfer learning feature extractor and a novel kNN-RNN-based sequential anomaly detector. The false alarm rate and threshold selection were analyzed mathematically. To address the common underuse of motion patterns and inconsistent datasets in current anomaly detection methods, ref. [69] proposed a system consisting of two parts: target tracking and suspicious activity. The system used ResNet tracking based on transfer learning to detect and track abnormal events in video surveillance in real time. This method avoided saturation performance when training deeper layers.

**Table 3 sensors-23-07442-t003:** Summary of online anomaly detection methods based on technical approaches.

Reference	Method	Architecture	Remarks
Ouyang et al. [53]	Deep learning	Randomly initialized multi-layer	The perceptron online optimization method and incremental learner were proposed.
Ullah et al. [55]		Vision transformer	A visual transformer with a sequential learning model and an attention mechanism was used.
Doshi et al. [56]		Transfer learning	The proposed semantic-embedding-based video anomaly detection method and transfer learning were used in different surveillance scenarios.
Rendón-Segador et al. [57]		3D DenseNet, multi-head self-attention mechanism and bidirectional convolutional LSTM	Violence in videos was detected by encoding relevant spatio-temporal features.
Ullah et al. [58]		CNN and LSTM	A surveillance scene dataset for indoor and outdoor environments was presented.
Kim et al. [59]		KCF and 3DResNet	An architecture that can extract pedestrian information in real time and detect abnormal behavior was used.
Mehmood et al. [60]		CNN	Proposing a lightweight framework that utilizes convolutional neural networks for training.
Lin et al. [61]		3DCNN and C3DGAN	Combined with 3D CNN and non-local mechanism, the synthetic data were converted into realistic video sequences through an adaptive recurrent 3D GAN.
Fang et al. [62]		CNN	Based on the improved yolov3 algorithm, the detection optimization was carried out using the frame-alternating two-thread method.
Direkoglu et al. [63]		MII and CNN	A new concept of MII was formed and combined with CNN for global abnormal crowd behavior detection.
Doshi et al. [64]	Transfer learning	KNN and KMeans clustering	A framework was proposed to determine the occurrence time of exceptions.
Doshi et al. [65]		Any-shot learning	A new framework with continuous learning capability was proposed for anomaly detection of statistical arbitrary shot sequences.
Doshi et al. [66]		Continual and few-shot learning	A continual learning algorithm was proposed.
Doshi et al. [67]		Multi-task learning	Proposing a multi-task learning framework and introducing a deep metric learning method based on semantic embedding.
Doshi et al. [68]		MOVAD	Combining an interpretable transfer learning feature extractor with a novel kNN-RNN sequential anomaly detector.
Kale et al. [69]		ResNet and DML	Transfer-learning-based ResNet object tracking and DML methods were proposed.

In the above research works, transfer learning was applied to online anomaly detection, but it was only limited to combining with neural networks, feature extractors and object trackers. So, combining transfer learning with more models and methods to build sustainable learning anomaly detection architectures is a focus of future research.

### 4.3. Other Methods

Instead of the above methods, as shown in Table 4, the use of Gaussian process regression models for anomaly detection in video surveillance was proposed by Li et al. [70]. Adaptive modeling of behavior patterns and anomaly detection in the current frame were achieved through online clustering algorithms and the utilization of supplementary information from previous frames. A cascade classifier for anomaly detection and localization was researched in [71]. It combined optical flow based on motion features and gradient-based appearance anomaly detection, thus simultaneously considering appearance and motion cues of video sequences and addressing the limitation of existing methods that only perform anomaly detection in the spatial dimension. Reference [72] introduced a method for automatic detection of suspicious behavior using CCTV video streams. The proposed method combined attention to motion amplitude and a gradient and used motion reactivity features extracted from optical flow to construct a suspicious behavior detection method based on a temporal saliency map. The effectiveness of the proposed method was verified by testing it on video clips containing suspicious behaviors. The framework of extracting a set of compact features based on foreground occupancy and optical flow information is a traditional approach for online anomaly detection [73]. This framework used an inference mechanism that evaluates a compact feature set to detect abnormal events through Gaussian mixture models, Markov chains and bags of words. However, the optical flow model thresholds and foreground occupancy model thresholds of this method were artificially set and did not automatically adapt to environmental transformations. Some researchers have proposed encoding spatio-temporal support regions with binary features instead of commonly using double precision features, combined foreground occupancy features and accurately classified events by relying on multiple low-complexity probability models to achieve online anomaly detection [74,75]. In [76], the authors combined the spatio-temporal gradient model with foreground detection to extract the significant spatio-temporal features on the foreground objects of video sequences, and then they used a sparse combinatorial learning algorithm to build an abnormal event detection model for anomaly detection. In order to be able to gradually learn new event samples from a video and detect anomalies in the video online based on the distance to the model and the relative frequency of each event class, a nonparametric hierarchical event model was proposed [77]. This method could lead to false alarms if the mesh was not properly delineated, localizing tiny human limbs or inaccurate optical flow estimation in crowded crowds. Feng et al. described a real-time online crowd behavior detection algorithm for video sequences [78]. The proposed algorithm was mainly based on a combination of visual feature extraction and image segmentation, which did not require an artificial neural network training phase to detect anomalies. The use of background subtraction in this method could make the algorithm less sensitive to rapid changes. The video abnormal event detection method based on multi-scale pyramid grid templates (MPGTs) proposed in [79] was innovative in multi-scale features, convolution operation and anomaly detection result fusion. Through the application of these methods, the accuracy and efficiency of video abnormal event detection can be improved.

### 4.4. Comparison

#### 4.4.1. Comparison of Advantages and Disadvantages of Different Methods

This paper focuses on the basic principles of distinguishing normal video from abnormal video and classifies online video abnormality detection methods into supervised, unsupervised, weakly supervised, deep learning, transfer learning and traditional methods based on comparison and summary as shown in Table 5.

The advantage of supervised learning in online video anomaly detection is that it can use labeled data to improve model performance, but it does not perform well in complex scenes. Unsupervised learning methods can effectively reduce false alarms and missed detection caused by model aging and threshold aging and avoid the problem of concept drift, but their performance will degrade in complex scenarios. Semi-supervised learning and weakly supervised learning methods can allow anomaly detection in real-time scenarios, but it is difficult to solve the problem of online constraints and weakly supervised settings simultaneously, and they are computationally expensive. Deep learning methods have a powerful learning ability and can extract more representative feature information from data, but they are highly dependent on data and have high computational complexity. The detection effect is good for video data with fewer anomalies, while the performance may be reduced in the case of a small number of datasets. Transfer learning methods can significantly reduce the computational complexity of the training and detection phases, but they have certain limitations when combined with the model. The method based on optical flow features is sensitive to moving objects, can extract speed and direction information, and has a good detection effect on data videos with few people or optical flows. Although the method based on binary features can reduce the frame processing time and improve the operation speed, the feature extraction will lose some important information, resulting in poor performance. In summary, different methods have advantages and disadvantages in different scenarios, and it is necessary to choose a suitable method according to the specific situation.

#### 4.4.2. Performance Comparison of Methods

Table 6 shows the performance metrics of these methods on representative scene datasets such as UCSD, ShanghaiTech, CUHK Avenue, UCF-crime, UMN and DOTA.

In general, the anomaly detection effect on the ShanghaiTech dataset is the best, and the anomaly detection effect on the UCSD Ped2 dataset is the second best. At the same time, the continuous learning regression method is used to obtain high detection accuracy on this dataset. This means that continuous learning techniques can help a model to continuously adapt to new data and improve the accuracy of the model when running for a long time. In addition, the better performance on the UCSD Ped1 dataset may be due to the fact that the scenes in this dataset are relatively simple and can be easily captured by the model for anomalies. In contrast, the AUC of the existing methods is generally low on the UCF-crime and DOTA datasets, probably because the scenes of these datasets are relatively complex, with more variations and noise. However, the UMN dataset is relatively less used. It is suggested that we need to consider the complexity and diversity of scenes when applying video anomaly detection methods and choose appropriate methods and datasets for research.

## 5. Summary and Outlook

In this paper, the research progress of online video anomaly detection technology is discussed in detail according to the method of example-based learning and technology. In addition, different datasets are presented, along with important anomaly details. At the same time, this paper also compares and analyzes the performance of the current mainstream algorithms according to various dataset evaluation criteria.

Current research on online methods in video anomaly detection faces the following main challenges and issues:(1)The models have a strong dependency on the task scenario and are difficult to directly transfer. The models only perform well on the datasets described in the paper, and the feature structure of the data is difficult to change due to strict requirements on some data formats (input size, color channels, text format, etc.); the trained feature extractors cannot be easily transferred to other tasks. Although these problems are mitigated by the “pre-training + fine-tuning” method, the transferability of the model needs to be further enhanced. Therefore, improving the transfer capability of models in online video anomaly detection is a problem worth investigating.(2)A real-time anomaly detection framework was built. Although most of the current methods can achieve high accuracy in anomaly detection, they cannot be deployed in real time. One of the key reasons is that the time cost of extracting effective features in the video is too high. From the perspective of practical applications, detecting anomalies in time and accurately can effectively reduce the loss caused by abnormal events. Therefore, it is necessary to design new methods for efficient video data preprocessing and feature extraction in the future, so as to break through the limitation of processing speed, so that more systems can be used for real-time detection scenarios.(3)A sustainable learning architecture for anomaly detection should be built. With prolonged monitoring, abnormal events are diverse and variable, and learned abnormal events may become normal events, making it difficult to identify abnormal behavior without expert interaction. Existing feature learning models trained offline are unable to adapt to such changes. Therefore, more online learning models with automatic update capability are needed to continuously update the anomaly detection knowledge base and improve the accuracy of detection in a continuous learning process.(4)The ability to locate the occurrence time of abnormal events should be enhanced. In video anomaly detection, it is necessary not only to detect and locate the abnormal event itself, but also to know the time when the anomaly occurs. The methods of temporal modeling and event analysis can be further explored to better understand the relationship and dynamic development of events in videos. This may include the use of methods such as spatio-temporal attention mechanisms, time-series models and event graphs.(5)Online and offline learning should be bridged with transformers. In practice, some decision tasks are not achievable outside of an online framework, and some environments are open-ended, meaning that strategies must be constantly adapted to handle tasks not seen during online interactions. Therefore, we believe that bridging online and offline learning is necessary. However, most of the current research advances based on decision transformers have focused on offline learning frameworks. Similar to offline reinforcement learning algorithms, distributional variation in online fine-tuning still exists; hence, there is a need for a special design of a decision transformer to address related issues. How to train an online decision transformer from scratch is a problem worth investigating.(6)A more appropriate online anomaly evaluation system should be found to make the evaluation of abnormal behavior more objective and comprehensive. The current evaluation metrics for video anomaly detection are mainly the area under the ROC curve (AUC) and the equal error rate (EER), but due to the unbalanced and heterogeneous nature of the data in the video anomaly detection datasets, using the equal error rate to evaluate the anomaly detection results will lead to large errors in practical application scenarios. The use of AUC to evaluate a model from frame-level and pixel-level criteria is also not very representative of the overall performance of the model. Therefore, it is important to establish a comprehensive and robust evaluation system for online video anomaly detection.

## Figures and Tables

**Figure 1 sensors-23-07442-f001:**
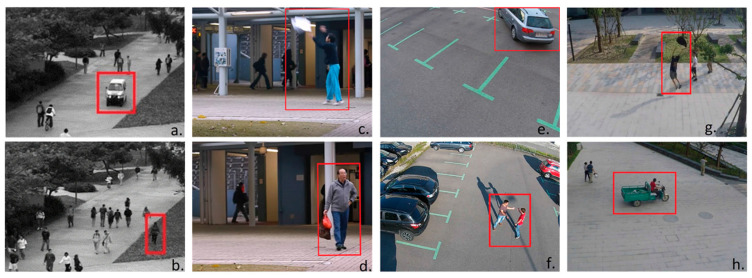
Anomaly detection in video surveillance scenes. (**a**) A truck moving on the footpath (UCSD Dataset). (**b**) Pedestrian walking on a lawn (UCSD Dataset). (**c**) A person throwing an object (Avenue). (**d**) A person carrying a suspicious bag (Avenue). (**e**) Incorrect parking of vehicle (MDVD). (**f**) People fighting (MDVD). (**g**) A person catching a bag (ShanghaiTech). (**h**) A vehicle moving on the footpath (ShanghaiTech) [5].

**Figure 2 sensors-23-07442-f002:**
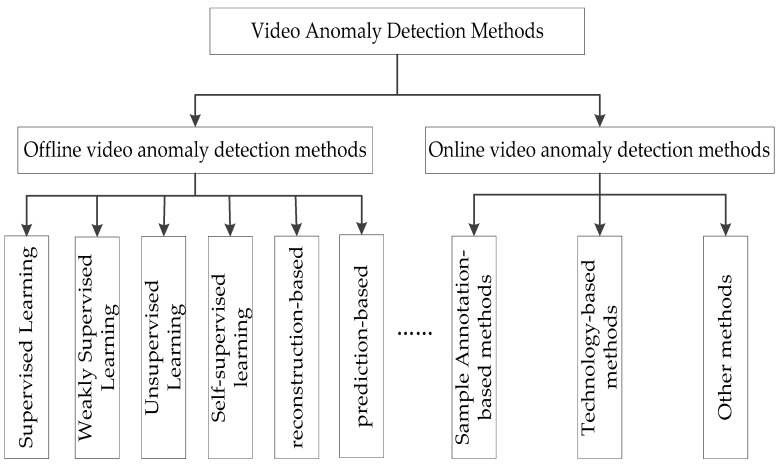
Video anomaly detection methods.

**Table 1 sensors-23-07442-t001:** Anomaly detection datasets for common scenarios.

Datasets	Labeling	Resolution	Type of Exception	Duration
UMN	Frame-Level	320 × 240	Escape, panic	27 min
UCSD Ped1	Pixel-Level and Frame-Level	238 × 158	Cycling, small vehicles	10 min
UCSD Ped2	Pixel-Level and Frame-Level	240 × 360	Cycling, small vehicles	10 min
CUHK Avenue	Frame-Level	640 × 360	Running, throwing objects, or walking in the wrong direction	30 min
ShanghaiTech	Frame-Level	856 × 480	Cycling, pushing prams, climbing over railings, running	3.01 h
UCF-Crime	Frame-Level	320 × 240	Abuse, arrest, arson, assault, road accidents, burglary, explosions, fighting, robbery, shooting, theft, shoplifting and vandalism	128 h
DOTA	Frame-Level	1280 × 720	Pedestrians, cars, trucks, buses, motorbikes, bicycles and riders	35 min

**Table 4 sensors-23-07442-t004:** Summary of other methods.

Reference	Method	Architecture	Remarks
Li et al. [70]	Other methods	Online clustering algorithm	An online clustering algorithm was used to construct the basic behavior model.
Li et al. [71]		ST-CAAE	A cuboid-patch-based spatio-temporal cascading auto-encoder (ST-CAAE) was proposed to improve the speed of anomaly detection in the next stage.
Cheoi et al. [72]		Optical flow and gradient features	A method for automatic detection of suspicious behavior using CCTV video streams was investigated.
Leyva et al. [73]		Foreground occupancy and optical flow functions	The joint response of the local spatio-temporal neighborhood model was considered to improve the detection accuracy.
Leyva et al. [74]		Binary features	Binary features were introduced to detect abnormal events in videos.
Leyva et al. [75]		Binary features	It used binary features to encode motion information and for low-complexity probabilistic model detection.
Geng et al. [76]		Sparse ensemble learning algorithms	Real-time abnormal event detection was realized by a sparse combination learning algorithm.
Feng et al. [77]		Incremental learning machine	A hierarchical event model for online anomaly detection in surveillance video was proposed.
Pennisi et al. [78]		Visual feature extraction and Image segmentation	A statistical analysis method combining feature detection and image segmentation was proposed.
Li et al. [79]		Particle filtering	A new particle prediction model and a weighted likelihood model were proposed.

**Table 5 sensors-23-07442-t005:** Advantages and disadvantages of different types of abnormal event detection methods.

Method Categories		Basis of Judgement	Advantages	Disadvantages
Methods based on sample annotation	Supervised learning	The optimal model is trained by training samples and applied to new data and output results so that the model has predictive ability.	Effectively use the information of data annotation to improve the prediction performance of the model.	Exception types are more complex, with poor performance and longer latency in complex scenarios.
	Unsupervised learning	The unlabeled training dataset needs to be analyzed to understand the statistical regularities between samples.	It reduces false alarms and missed detections caused by model threshold aging and avoids concept drift in unstable environments.	Lack of ability to automatically update and distinguish anomalies online based on scene changes.
	Weakly supervised learning	Incomplete, inexact or imprecise labeling information is called weak supervision and is widely used in label learning.	Real-time online detection of anomalies without additional buffer time.	The single-stage framework is not suitable for solving multi-stage problems and is computationally expensive.
Methods based on technical approach	Deep learning	A deep neural network is constructed, and a large number of sample data are used as input to obtain the model and learn its internal laws and representation levels.	Good portability is directly proportional to the amount of data, and deep learning is highly data-dependent.	There are challenges and high computational complexity in dealing with online learning, noise and concept drift.
	Transfer learning	Using the similarity between data, tasks or models, a model that has learned in an old domain is applied to a new domain to solve a new problem.	The computational complexity of the training and detection phases can be significantly reduced.	The combination with the model has certain limitations.
Other methods	Optical flow features	When the object is moving, the brightness pattern of its corresponding point on the image is also moving.	Optical flow expresses the information of image changes and object motion, which is suitable for detecting abnormal behavior activities in videos.	When changes in optical flow are not obvious, abnormal behavior cannot be accurately detected.
	Binary features	A number of point pairs are randomly selected around a feature point, and their gray values are combined into a binary string, which is used as the feature descriptor of the feature point.	Frame processing time can be reduced, greatly increasing the speed of computing.	Some important feature information can be lost, resulting in a loss of accuracy.

**Table 6 sensors-23-07442-t006:** Performance comparison of different online video anomaly detection methods.

Author/Year	Model Structures	Frame Levels AUC/EER/%						
		UCSD PED1	UCSD PED2	ShanghaiTech	CUHK Avenue	UCF-Crime	UMN	DOTA
Pennisi [78]/2016	Statistic Analysis	-/-	-/-	-/-	-/-	-/-	95.0/-	-/-
Chaker [39]/2017	SNN	-/-	87.9/-	-/-	-/-	-/-	-/-	-/-
Sun [43]/2017	Online GNG	93.75/-	94.09/-	-/-	-/-	-/-	-/-	-/-
Leyva [75]/2017	Binary Features	-/-	-/-	-/-	-/-	-/-	88.3/-	-/-
Sultani [12]/2018	MIL	-/-	-/-	-/-	75.41/-	-/-	-/-	-/-
Leyva [76]/2018	Binary Features	-/48.1	-/38.4	-/-	-/-	-/-	-/-	-/-
Doshi [65]/2020	Continual Learning	-/-	97.8/-	71.62/-	86.4/-	-/-	-/-	-/-
Li [71]/2020	ST-CAAE	-/-	87.1/-	-/-	-/-	-/-	-/-	-/-
Doshi [35]/2021	GAN	-/-	97.2/-	70.9/-	-/-	-/-	-/-	-/-
Majhi [50]/2021	Cov-DAM	-/-	-/-	88.22/-	-/-	82.67/-	-/-	-/-
	Dist-DAM	-/-	-/-	88.86/-	-/-	82.57/-	-/-	-/-
Huang [52]/2022	TTFA and SDFE	-/-	-/-	98.06/-	-/-	84.04/-	-/-	-/-
Nawaratne [44]/2022	ISTL	75.2/29.8	91.1/8.9	-/-	79.8/29.2	-/-	-/-	-/-
Liu [51]/2022	Hybrid Conv-Transformer	-/-	-/-	-/-	-/-	85.18/-	-/-	-/-
Doshi [67]/2022	DNN and Continual Learning	-/-	95.6/-	70.12/-	86.4/-	-/-	-/-	-/-
Doshi [68]/2022	Continual Learning	-/-	97.2/-	73.62/-	88.7/-	-/-	-/-	-/-
Rossi [37]/2023	VST	-/-	-/-	-/-	82.11/-	-/-	-/-	82.1/-
Yao [13]/2023	FOL-Ensemble	-/-	-/-	-/-	-/-	-/-	-/-	73.0/-
Ouyang [53]/2023	MLP	-/-	96.5/-	83.1/-	90.2/-	-/-	-/-	-/-
Ullah [55]/2023	3D-CNN	-/-	-/-	-/-	-/-	51.0/-	-/-	-/-
Doshi [56]/2023	Continual Learning	-/-	-/-	68.9/-	79.0/-	-/-	-/-	-/-

## Data Availability

There are no available data to be stated.

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
