# Peer review of "Online Video Anomaly Detection"

_sensors, 2023, doi:10.3390/s23177442_

Round 1
Reviewer 1 Report
The paper presents an analysis on current video anomaly detection trends and their effectiveness in current and future real-time applications. The authors presented the main branches of underlying technologies that are applicable for both online (real-time) and offline video action detection and also provided ad-hoc comparisons. The primary limitations of current technological trends of finding out points of interest in real-time from video surveillance are presented.
Overall the quality of the content of the paper is acceptable with few grammatical errors (please see yellow highlights). The paper brought together different ideas and research in the field of computer vision and machine learning that are used in the video anomaly detection application domain. Thus, this paper provides a starting point for future researchers to learn about current trends in video-based actions and related anomaly detections for the greater purpose of video surveillance systems.

It is recommended that the authors employ the services of a native English speaker to correct the english language errors.
Reviewer 2 Report
The major issue for this survey paper is that the comparisons of different methodologies are too shallow. More figures and tables could help Other than that, I recommend the authors highlight their own discoveries rather than just rephrase and submit.
The language is understandable, but the depth of the paper is hardly sufficient to get published.
Reviewer 3 Report
The paper is not about sensors. It should be rejected because out of scope.
Moreover:
- The paper lacks of technical quality. It does not review deeply the surveyed papers
- The paper should include a performance test comparison including the surveyed papers.
The English Language is good
Reviewer 4 Report
Overall, the paper's presentation is fine. However, there is a great number of studies in the area that have not been included. I am producing a list of significant works below that must be addressed. Yet, it is still not an all-inclusive list and I'm providing it to highlight the fact that the search methodology of the paper lacks maturity.
Further, by a simple database search I could locate yet another set of tens of studies in the specific area of this paper which were published in 2022 and 2023.
Authors need to have a more comprehensive approach and a well-defined method that ensures to include all representative studies in the topic area.
Studies I found (more need to be searched and included)
LightAnomalyNet: A Lightweight Framework for Efficient Abnormal Behavior Detection (https://doi.org/10.3390/s21248501)
Temporal Saliency-Based Suspicious Behavior Pattern Detection (https://doi.org/10.3390/app10031020)
ViolenceNet: Dense Multi-Head Self-Attention with Bidirectional Convolutional LSTM for Detecting Violence (https://doi.org/10.3390/electronics10131601)
Real-Time Surveillance System for Analyzing Abnormal Behavior of Pedestrians (https://doi.org/10.3390/app11136153)
An intelligent system for complex violence pattern analysis and detection (https://doi.org/10.1002/int.22537)
Examination of Abnormal Behavior Detection Based on Improved YOLOv3 (https://doi.org/10.3390/electronics10020197)
Deep learning and handcrafted features for one-class anomaly detection in UAV video (https://doi.org/10.1007/s11042-020-09774-w)
Abnormal Crowd Behavior Detection Using Motion Information Images and Convolutional Neural Networks (https://doi.org/10.1109/ACCESS.2020.2990355)
Learning to detect anomaly events in crowd scenes from synthetic data (https://doi.org/10.1016/j.neucom.2021.01.031)
Spatial-temporal cascade autoencoder for video anomaly detection in crowded scenes (https://doi.org/10.1109/TMM.2020.2984093)
Round 2
Reviewer 3 Report
The paper is still out of scope. There is nothing related with sensors.
Moreover, authors have not addressed my comments:
- The paper lacks of technical quality. It does not review deeply the surveyed papers
- The paper should include a performance test comparison including the surveyed papers.
Reviewer 4 Report
NA
